# Functional Annotations of Paralogs: A Blessing and a Curse

**DOI:** 10.3390/life6030039

**Published:** 2016-09-08

**Authors:** Rémi Zallot, Katherine J. Harrison, Bryan Kolaczkowski, Valérie de Crécy-Lagard

**Affiliations:** Department of Microbiology and Cell Science, Institute of Food and Agricultural Sciences, University of Florida, Gainesville, FL 32611, USA; remizallot@ufl.edu (R.Z.); katherinejh@ufl.edu (K.J.H.); bryank@ufl.edu (B.K.)

**Keywords:** paralog, ortholog, homolog, annotation, genome context, phylogeny, signature motif, genome, multigene families

## Abstract

Gene duplication followed by mutation is a classic mechanism of neofunctionalization, producing gene families with functional diversity. In some cases, a single point mutation is sufficient to change the substrate specificity and/or the chemistry performed by an enzyme, making it difficult to accurately separate enzymes with identical functions from homologs with different functions. Because sequence similarity is often used as a basis for assigning functional annotations to genes, non-isofunctional gene families pose a great challenge for genome annotation pipelines. Here we describe how integrating evolutionary and functional information such as genome context, phylogeny, metabolic reconstruction and signature motifs may be required to correctly annotate multifunctional families. These integrative analyses can also lead to the discovery of novel gene functions, as hints from specific subgroups can guide the functional characterization of other members of the family. We demonstrate how careful manual curation processes using comparative genomics can disambiguate subgroups within large multifunctional families and discover their functions. We present the COG0720 protein family as a case study. We also discuss strategies to automate this process to improve the accuracy of genome functional annotation pipelines.

## 1. Introduction

The advent of low-cost, high-throughput sequencing technologies has created the global capacity to generate complete genome sequences at an unprecedented rate, with tens of thousands of organisms currently at various stages in genome sequencing pipelines [1,2,3]. This rate is expected to continue increasing, at least for the foreseeable future, as technologies continue to improve. This genome sequencing “revolution” has profound implications. Classically, analysis methodologies focused heavily on rigorous small-scale experimentation and statistical approaches for boosting power to detect differences in small data sets [4,5,6]. The deluge of genome sequence data makes small-scale experimental approaches impractical, while the coincident increase in computational power has made high-throughput bioinformatics analyses an attractive alternative [7,8,9]. Out of necessity, biologists are relying increasingly on bioinformatics-based predictions to generate information [10].

Although the development of efficient algorithms for identifying protein-coding sequences (CDSs) is still an active area of bioinformatics research, “gene finding” has improved greatly in recent years, particularly for prokaryotes [11,12,13,14,15] (even if the calling of start sites is still often problematic [16]). However, the functional annotation of CDSs is particularly difficult to automate. Current state-of-the-art functional annotation pipelines integrate multiple types of evidence [17,18,19], but unfortunately the quality of functional annotations remains generally poor [20,21,22,23] and is highly dependent on resource-intensive manual curation [24,25].

Early pioneers in function annotation were quick to identify potential problems with large-scale annotation efforts [26,27,28], and misannotation is a growing concern among the general research community, as misannotated genes can have a “ripple effect” impacting diverse areas of biological inquiry [29,30,31]. By different measures, 10%–25% of functional calls are wrong, even in very small bacterial genomes [32]. High-throughput functional calls can be incorrect due to a variety of factors [33,34], but the most common errors (>85%) are over-annotations, in which a gene is given a specific but incorrect function [21,32,35]. Once made, functional annotation errors can be difficult to correct in large sequence databases and can often “propagate” themselves to newly sequenced genomes through “(mis)annotation transfer” [36,37,38,39].

Multigene families—defined as having more than one member of the family in a given genome—can be notoriously difficult to correctly annotate, primarily due to over-annotation problems [21]. Large multigene families are also a potential goldmine for discovering new functions, because many of the enzymes within a family are likely to perform chemically similar reactions on similar substrates, reducing the functional space to explore in comparison with totally unknown enzymes (see Table 1 for examples). Overcoming the challenges of annotating multigene families and exploiting their potential for discovering new gene functions will likely require integrating large-scale and small-scale annotation methods. To date, the majority of genome-scale functional annotation pipelines have been developed in isolation from small-scale annotation efforts or experimental validation. While large-scale approaches are inherently limited due to the need for computational efficiency and generalizability, many smaller-scale efforts are combining advanced bioinformatics tools with experimental validation to disambiguate multigene families and are producing high quality functional annotations and discovering novel or missing functions in the process [40,41,42,43]. In our view, “closing the loop” between these different approaches has a tremendous potential to improve the quality of functional annotations stored in major online databases. A recent call for functional microbiologists to be more involved in the genome annotation process stresses this point [44].

In this work, we argue that focusing on individual genomes that have readily identifiable paralogs and examining similarities and differences in the corresponding gene neighborhoods is an efficient way to identify and separate similar genes with different functions. We propose an annotation workflow that “bootstraps” information gained by identifying specific genomes harboring functionally differentiated paralogs to potentially improve annotations of related genes in other genomes (Figure 1). We expect this type of annotation workflow could both guide the experimentalist and improve automated annotation of large multigene families.

## 2. Protein Function and Evolution

Protein functional annotation can vary widely in the degree of precision and can be defined in multiple ways, including phenotype, cellular localization, ligand interactions or interaction partners [59,60]. Efforts to unify and standardize functional annotations using controlled hierarchical vocabularies—although controversial and incomplete—have provided an objective framework for functional annotation that can be computed automatically [61,62]. However, these approaches are only beginning to be incorporated into genome annotation pipelines, requiring older genomes to be re-annotated [63,64]. For the purpose of this study, we will use a very strict definition of function in which both the molecular function and its functional context have been elucidated (sometimes called a two-dimensional annotation [65]). For an enzyme, this would be the Enzyme Commission number (EC number) and the biological pathway the enzyme participates in.

As it is impossible to experimentally verify the function of every newly sequenced gene, researchers have sought to elucidate the “evolutionary rules” governing when it is—and when it is not—appropriate to “transfer” specific functional information from a gene of known function to a related gene whose function is not known. We would expect genes with very similar sequences to have similar functions, while genes with more dissimilar sequences are more likely to have different functions. Indeed, the earliest functional annotation algorithms simply used sequence similarity searches to predict protein function by “annotation transfer” [66,67], but these generally led to unacceptably high error rates [32,33]. The fact that gene duplication and speciation events followed by mutation can lead to functional changes means that proteins with high sequence similarity might not have the same function [68].

More sophisticated approaches attempt to identify specific evolutionary patterns that may correlate with functional conservation or divergence [69,70,71]. The evolutionary relationship between two genes in a protein family (i.e., “homologs”, genes that have descended from a common ancestor) can be broadly classified into two types: “orthologs” are two genes from different species that derived from a single gene in the last common ancestor of the species, while “paralogs” are two genes (within the same or different species) that derived from a single gene that was duplicated within the genome of some species [72].

Current thinking and empirical evidence supports the conclusion that, in general, orthologous genes have equivalent functions more often than paralogs [73]. This has led to numerous approaches for identifying groups of orthologs (e.g., OrthoMCL, OMA and eggNOG [74,75,76]) and transferring functional annotations among members of an ortholog group. Some researchers have gone so far as to “redefine” the term “orthologs” to mean “genes with equivalent functions” [77]. In agreement with R.A. Jensen [78], we feel that the “functional” definitions of “ortholog” and “paralog” are unwarranted and invite confusion, and we retain the original “evolutionary” definitions in this work. There are certainly examples of orthologs that have different functions in different species [78,79,80,81], and functional differentiation between orthologs from different species (particularly those separated by large evolutionary distances and/or major changes in lifestyle or habitat) may be common enough that simply transferring functional annotations among orthologs—without considering additional information—is likely to lead to potential errors [82].

It may be comforting to assume that the converse relationship may hold more universally: perhaps paralogous genes in the same species always have different functions. However, duplicate genes with equivalent functions can be retained to supply specific gene or protein dosage, providing a scenario in which paralogous genes have equivalent functions [27,83,84,85,86]. In multicellular eukaryotes, this can occur when gene regulatory functions are partitioned between daughter genes (i.e., the same protein is expressed by different genes in different tissues, cell types, or different periods of time/development). This regulatory split also occurs in bacteria [87,88]. Hence, orthology/paralogy—in and of itself—is insufficient as a universal means for dividing related genes into functional classes, although it can provide crucial information as part of a more comprehensive strategy.

## 3. Identifying Orthologs and Paralogs in Practice

In theory, identifying orthologs and paralogs with phylogenetic trees should be straightforward. “Ortholog” and “paralog” are evolutionary concepts that are well-defined from a phylogenetic perspective; given a gene family tree and a species tree that are both known with absolute certainty, orthologs and paralogs can be identified without error [89,90]. However, neither gene nor species trees are ever known with absolute certainty, and current gene-species tree reconciliation methods have limited capacity to incorporate phylogenetic uncertainty or complex realistic evolutionary scenarios [49,91,92,93,94]. Even sophisticated phylogenetic reconstruction methods can be subject to methodological bias under some conditions [95,96,97,98], and strongly supported errors in either the gene tree or the species tree can produce radical errors in ortholog/paralog identification [97]. Even if the phylogenies of extant genes and species could be reconstructed without error, certain patterns of differential gene losses can make paralogous genes erroneously look like orthologs (Figure 2).

Even with these caveats, recent phylogeny-based functional annotation methods have performed very well, compared to competing approaches [99,100]. Unfortunately, rigorous phylogenetic analysis does not scale well computationally, making it difficult to apply the most accurate tree reconstruction algorithms to large data sets [71,101]. Fast tree reconstruction algorithms do exist but are more prone to errors in both topology and confidence assessment than more computationally intensive methods [97,102,103,104]. Even with the dramatic recent speed-up of phylogenetic reconstruction algorithms, these methods are likely to remain too computationally expensive to deploy across the growing wealth of whole-genome sequence data [71,105].

Numerous approaches have been devised that use computationally efficient sequence similarity comparisons, often combined with other sources of information, to approximate phylogenetically based ortholog/paralog identification while improving throughput [106,107]. At their core, most of these approaches use patterns of pairwise sequence similarity among genes to infer patterns of orthology/paralogy. Historically, Koonin and co-workers were the first to implement such an approach with their Clusters of Orthologous Groups (COGs), based on best birectional BLAST hits [108]. Many other approaches to ortholog/paralog identification use pairwise sequence similarity as a starting point, differing mainly in how pairwise similarity networks are divided into ortholog/paralog groups [74,109,110]. For example, Sequence Similarity Networks (SNNs) have been used successfully to identify paralogs and isofunctional groups in large multigene families [111]. A major advantage of SSNs over traditional phylogenetic trees is that relationships between protein sequences are easier to visualize, allowing larger numbers of sequences to be examined, and it is generally easier to automatically analyze similarity networks to divide them into orthology groups.

It is well recognized that patterns of sequence similarity only weakly approximate phylogenetic relationships [71,112,113,114], so these approaches may be error-prone in orthology/paralogy inference. As shown in Table 1, some known paralogs have been erroneously grouped in the same COG, and some orthologs are erroneously split into different COGs. To estimate how frequently COG families contained paralogs, we analyzed the genomic distribution of the latest COG set that lists 4695 COGs across 711 prokaryotic and archaeal genomes (2003 COGs, 2014 update) [58]. This analysis revealed that only 12% of COGs have only one member in any given genome. For 88% of COGs, a genome could be found that encodes at least two paralogous members of the same COG, reinforcing that sequence similarity alone is insufficient to reliably group genes into sets of orthologs (Figure 1). Of course, if errors in orthology/paralogy inference by sequence similarity happen to correlate with the historical patterns of functional divergence among orthologs/paralogs, sequence similarity methods may perform surprisingly well at predicting function, albeit because they may be fortuitously “biased in the right direction” [114,115].

## 4. Lessons from Comparative Genomics

Comparing similarities and differences among genomes can be a powerful tool for functional annotation. Not long after the release of the first whole genome sequence, visionary pioneers like Koonin [116] and Overbeek [117,118] provided a framework for using comparative genomic methods to improve gene annotations. In general, the “comparative genomic framework” combines evolutionary relationships with additional information gleaned from whole-genome comparisons to improve predictions about similarities and differences in gene function.

A gene’s function is impacted by the specific metabolic network or subsystem within which it works as well as its role within that network or subsystem [65,119,120,121]. One of the underlying principles of the comparative genomic framework is that various types of information, such as physical clustering, gene fusions, co-regulation or regulons and phylogenetic co-distribution, can be used to reconstruct a gene’s potential role within a network or subsystem [122,123,124]. These association networks can be used to efficiently identify and annotate paralogs that have diverged in function, particularly in the case of prokaryote genomes, where genes operating in the same subsystem are often physically clustered. As the first dozens of bacterial genomes were sequenced, it became apparent that conserved physical clustering of genes (conserved genome context) in phylogenetically distant genomes is a strong suggestion of functional association [118,125,126,127]. The underlying causes can vary but include maintaining co-regulation of functionally related genes [128,129,130] and facilitating horizontal transfer of multigene functional units [131]. As more genomes have become available, physical clustering and gene fusions—which can also be considered a form of physical clustering [125,132]—have become a robust measure of functional association [127,133,134] used in various integrated annotation platforms (e.g., STRING [135], SEED [118,136], Microbesonline [137], Syntax [138], MaGE [139]).

Examples of successful separation and annotation of paralogs within multigene families by our group are given Table 1. In all cases, physical clustering analysis was key to the functional separation of the multigene family into putative isofunctional subgroups. These examples are drawn from families with only two to three subgroups, but physical clustering can also be useful for functionally characterizing more complex gene families [140,141]. For example, a single bacterial genome can contain dozens of individual genes from the Nudix hydrolase family. Members of this family are all pyrophosphatases [142] but can vary widely in preferred substrate and are very difficult to functionally separate by sequence similarity alone. However, substrate specificity can be accurately predicted by incorporating physical clustering information [71,105,142,143].

Combining phylogenetic or sequence similarity information with additional comparative-genomics information—gene clustering, protein-protein interactions, regulon membership, presence/absence of specific functional domains, structure, chemistry and “signature” sequence motifs—has produced some of the most effective examples of reliably separating functional groups within multigene families [43,106,144]. These approaches are extremely effective when performed on a single gene family, often in combination with experimental validation [145,146]. Although these integrative approaches to functional annotation currently rely on expert understanding of the specific gene family under examination and careful manual curation, recent advances have improved prospects for future complete automation of integrative function-prediction analyses, with impressive results. For example, the Structure-Function Linkage Database (SFLD) analyzes 12 superfamilies that cover over 300 functional families [147], and the CATH-Gene3D and FunFams resources encompass more than 2500 families [148,149]. These “middle level” resources fill an important gap between rigorous small-scale experimentation and efficient high-throughput annotation pipelines; they are likely to improve the accuracy of functional annotations for the gene families they cover but are not currently deployed at a whole-genome scale. Future expansion of these and similar resources that carefully integrate multiple data sources using sophisticated approaches that mimic the manual curation process are expected to dramatically improve the overall quality of gene functional annotation.

## 5. The COG0720 Case Study

The COG0720 family is a good example of how comparative genomics can differentiate paralogs, even in complicated cases with several functional divergence events. The founding member of COG0720 is 6-pyruvoyl-tetrahydropterin synthase (PTPS). This enzyme catalyzes the second step of tetrahydrobiopterin (BH4) biosynthesis. BH4 is a cofactor used by aromatic acid hydroxylases in animals and certain bacteria [150]. It is synthesized from guanosine triphosphate (GTP) in three enzymatic steps (Figure 3A). The first step is shared with the folate biosynthesis pathway and catalyzed by GTP cyclohydrolase I (FolE). PTPS (EC 4.6.1.10) then produces 6-pyruvoyl-tetrahydropterin (PTP) from dihydroneopterin-triphosphate (DHN-P3). The last step is catalyzed by sepiapterin reductase (SR) (Figure 3A). The first cloned and sequenced PTPS-encoding gene was from rat [151]. Some bacteria known to have a BH4 biosynthesis pathway, like *Synechococcus* sp. *PCC 7942*, have two PTPS homologs. PTPS-I (BLAST *e*-value 6 × 10^−31^ to the rat homolog) was shown not to be involved in BH4 synthesis, as deletion of the corresponding gene did not affect the level of synthesis of the cofactor [152], while PTPS-II (BLAST *e*-value 5 × 10^−20^ to the rat homolog) was shown to have canonical PTPS activity in vitro [152] (Figure 3A) (default BLAST parameters were used). This is a clear case of bacterial paralogs with different functions and also demonstrates that the best BLAST hit can be the wrong paralog.

Although COG0720 proteins are found in most bacteria, only ~50 are predicted to take part in the BH4 biosynthesis pathway based on literature review and co-occurrence with the SR protein (see “PTPS paralogs” subsystem http://pubseed.theseed.org//SubsysEditor.cgi?page=ShowSubsystem&subsystem=PTPS_paralogs) [119]. Hence, metabolic reconstruction could have flagged the 6-pyruvoyl-tetrahydropterin synthase annotation for most members of the COG0720 family as questionable. However, this correction did not occur, and most COG0720 genes were incorrectly annotated. In the initial COG database [116], COG0720 contained 55 proteins in 43 archaeal or bacterial genomes. Even though eight genomes encoded multiple COG0720 copies, all COG0720 members were annotated as 6-pyruvoyl-tetrahydropterin synthases. Our more recent analysis of the Joint Genome Institute (JGI) Integrated Microbial Genomes (IMG) database (Table 2) found that ~15 percent of bacterial genomes contain more than one COG0720 member. The COG0720 family is a clear case of over-annotation caused by transfer of annotations between functionally divergent paralogs.

Using different types of comparative genomics evidence, we were able to predict, and then experimentally verify, that at least two other COG0720 sub-families (PTPS-III and PTPS-I) perform different reactions in other pathways (Figure 3A) [55]. PTPS-III is part of the tetrahydrofolate pathway in *Plasmodium falciparum* and various bacteria [55,153,154]. In specific Archaea, members of the PTPS-III group are involved in tetrahydromethanopterin biosynthesis, but in these cases the substrate might be the monophosphate and not the triphosphate form [51]. PTPS-I is involved in the synthesis of archaeosine and queuosine in tRNA [55,155,156]. In this case, physical clustering evidence (Figure 3B) was sufficient to link the three PTPS gene families to their respective biochemical pathways.

Multiple alignments and structure comparisons allowed us to identify diagnostic signature motifs for the subfamilies (Figure 3C) [55]. Enzymes belonging to the PTPS-II subfamily harbor the motif CxxxxxHGH. The motif found in PTPS-III enzymes ExxHGH is not mutually exclusive with the one found in PTPS-I enzymes QueD (CxxxHGH), i.e., an enzyme with a CExxHGH can exist. Some organisms indeed have a PTPS-I/III hybrid version of the enzyme that can function in both folate and Q pathways [55]. In addition to the examples presented here, other subfamilies have been identified: PTPS-IV is involved in the synthesis of the archaeal cofactor F420 (de Crécy-Lagard and R. White unpublished); PTPS-VI has been validated as a folate enzyme [51]; and the function of PTPS-V is still unclear. Each of these subfamilies harbors specific signature motifs [55]. In summary, the functional complexity of the COG0720 family—which appears only separable using a combination of comparative techniques—makes it a good example to test automated annotation pipelines.

## 6. Current Methods for Paralog Annotation in Genome Annotation Pipelines

The above discussion suggests that careful methods for separating multigene families into functional subgroups have become highly effective. However, these robust approaches are not regularly incorporated into automated genome annotation pipelines, so the “best” methods are not being used to annotate most genes. Indeed, in most cases, after a given CDS has been predicted, it is aligned using sequence similarity to either a reference genome or to protein sequences available in large databases such as Uniprot. If a high scoring homolog is identified, the annotation is typically transferred, often without any significant additional analysis [67]. This process is far from perfect, as mentioned above and discussed in detail by Richardson et al. [67]. Issues include the use of references with outdated or misspelled annotations, inconsistent annotations among close orthologs, protein fusions and, of course, paralogs.

A recent improvement in annotation pipelines has been to include family relationships (e.g., OrthoMCL, CDD, COG, EggNOG, Pfam, TigerFam, FigFam) as part of the annotation [67]. This is very useful for identifying fusion proteins that are a well-known misannotation problem [67,132]. However, unless domain families have been specifically designed to separate paralogs (such as FigFam, used by the RAST pipeline), domain annotations do not disambiguate paralogs; on the contrary, domain-family annotations typically group paralogs together. We evaluated if and how paralogs were separated in the most popular genome annotation pipelines, including: IMG [17,157,158], RAST [13,159,160], NCBI prokaryotic annotation [161], EnsemblBacteria [162] and MaGE [139]. We specifically examined two COG0720 proteins from *Pyrococcus furiosis* DSM 3638: PF0219 and PF1278. The first is a (preQ0) synthesis enzyme (PTPS-I) and the second a folate synthesis enzyme (PTPS-III).

The NCBI Prokaryotic Genome Annotation Pipeline relies on protein cluster membership to transfer functional annotations [161,163]. This method is not designed to separate paralogs without additional curation. We found that PF0219 was correctly annotated, but PF1278 was not, and it remains hypothetical (Table 3). IMG uses a combination of methods to assign IMG terms as a controlled vocabulary for describing functional roles [164,165], but IMG relies on BLAST or family relationships that do not explicitly separate paralogs. Interestingly, IMG has developed a series of integrated tools to identify identical functional annotations within a genome and flag paralogs [166]. However, these tools are not directly used in the annotation pipeline and are not easily accessible from a given gene page through the IMG web browser interface. Many genomes have been re-annotated in IMG [157], but if PF0219 was correctly annotated as a QueD enzyme in the first annotation, the re-annotation introduced a functional ambiguity that obscured the PF0219 annotation. For PF1278, the function was not called in the first IMG annotation, and it was miscalled in the re-annotation (Table 3).

RAST uses a curated set of protein families (FigFams) as a source of annotations for submitted sequences. FigFams are defined as sets of isofunctional homologs based on sequence similarity and conserved genomic context across closely related genomes [159,167]. This family definition should allow separation of paralogs, and the two test proteins were indeed correctly called (Table 3). We should note that, because of our long association with the SEED database that is the basis for many RAST/PATRIC annotations, these genes were actually manually curated by our group. If the initial curation of a few members of the FigFam is not correctly done, the paralog families will not be separated.

MaGe (Magnifiying Genomes) annotates proteins based on sequence similarity searches against non-redundant protein sequences, protein family identification and using HAMAP (High quality Automated and Manual Annotation Proteins) profiles [168]. Gene contexts are said to be taken into account in annotation calls [139], so in theory this should allow paralog separation. However, it does not appear that this information is integrated at the level of functional annotation, as neither of the two test proteins was correctly annotated in MaGe (Table 3). Finally, Ensembl Genomes uses the InterProScan 5 pipeline for protein annotation [169,170], which primarily uses HMM (Hidden Markov Model) signatures for identifying protein families [170]. If paralogs have not been separated by the HMM signatures beforehand, they will not be separated in any new annotation. In addition, the Ensembl platform provides a gene-oriented phylogenetic resource that should allow paralog identification (see [171]). However, it is not available for every gene entry and does not appear to be used for annotation purposes, as neither of the two test proteins was correctly annotated in Ensembl (Table 3).

In summary, we found a major discrepancy between the tools available in large integrated databases, which should in theory be capable of disambiguating or at least flagging paralogs, and the actual annotation pipelines, which rarely appear to take advantage of these tools to transfer functional annotations. In practice, large-scale functional annotation is still mainly based on similarity scores alone. The identification of paralogs remains primarily a low-throughput manual process.

## 7. Integration of Tools for Paralog Separation in a Workflow

As discussed above, paralogs within large multigene families remain a potentially major source of errors in high-throughput gene annotation pipelines. A “stop-gap” first measure for mitigating the effects of paralogous genes on annotation quality could be to simply flag potential paralogs and integrate this flagging into annotation confidence scores [18]. If a gene has a homolog in the same genome, or even if it is part of a family that has several members in another genome, the confidence in its functional annotation should be reduced. This would introduce a level of “healthy skepticism” into predicted functional annotations that is lacking in most cases. This first step could be implemented efficiently and easily incorporated into existing annotation pipelines, as data on the distribution of gene families across genomes is already available in most integrated databases such as PATRIC, IMG or MaGE. While flagging potential paralogs is an appropriate first measure, the ultimate goal here is to examine the extent to which tools for rigorously separating multigene families into isofunctional subgroups could be integrated into high-throughput annotation pipelines.

Information from the highly curated SFLD [147] or CATH-Gene3D platforms [148,149] are considered the current “state of the art” in subfamily separation and classification. The COG0720 family is not currently covered by SFLD, but it is in CATH (CATH ID: 3.30.479.10). However, the CATH pipeline was not able to correctly separate PTPS subfamilies, even if it did capture the chemistry for two of them, PTPS-I and PTPS-II. That said, for the families covered, information from SFLD and CATH-Gene3D has the potential to improve gene functional annotations and should therefore be incorporated into annotation pipelines. Even if only some gene family annotations are expected to be improved by incorporation of manually curated database information, incorporating this information should help mitigate misannotation transfer errors and allow more generalizable—but computationally expensive—methods to be applied in the more difficult cases for which they are needed.

We also tested whether SSNs and Genome Neighborhood Networks (GNNs) developed by EFI [111] could separate the different COG0720 subgroups. We created an SSN from the COG0720 proteins extracted from the “PTPS paralogs” subsystem, a set of sequences for which the annotations have been verified. Increasing stringency allowed good separation of the different PTPS subfamilies into clusters, but the alignment score threshold required to fully separate the functional groups was so high that the same subfamily is also split into different clusters (Figure 4A). This problem persisted when we selected a more stringent set of starting sequences by examining only subsystem sequences from genomes that contained more than one COG0720 family member (Figure 4B). As this example demonstrates, in some “real-world” cases, sequence similarity may not be sufficient to identify all isofunctional groups within a multigene family without also breaking some isofunctional groups into multiple clusters. We then built the corresponding GNNs, as these might allow the regrouping of clusters based on common gene neighborhoods (Figure 7). The GNN generated did find neighbors that are relevant to the functions of the PTPS subfamilies. For example, we observed the expected physical clustering between PTPS-I and QueC, QueF and FolE. However, proteins of the same family were found in the neighborhood of proteins from different similarity clusters and more generally the noisiness of the GNN results make them difficult to interpret. If nothing were known about the family being analyzed, determining functional groups precisely using this data would be a challenge.

In summary, the EFI network tools were able to create networks that separate the COG0720 subfamilies. However, their analysis requires prior knowledge of the families as well as trial and error in order to provide accurate and reliable information about protein function. Noise-reduction techniques and other similar statistical tools could be developed to make these methods more useful for the average user or easier to integrate into automated annotation pipelines, but these additional analyses would need to be validated prior to deployment on a large scale.

None of the automated functional annotation pipelines were able to correctly separate COG0720 paralogs, whereas these paralogs were easily identified manually using comparative genomic tools. This suggests that an automated workflow inspired by our manual annotation strategy might be appropriate for separating challenging paralog groups (Figure 5). The key idea in this workflow is to focus initial analyses only on those genomes in which paralogs can be easily identified. Simply, any single genome containing at least two members of a protein family must encode paralogs, although the paralogs could perform the same or different functions. Once paralogs have been identified in a single genome, physical clustering by gene neighborhood can be used to group paralogs likely to have similar functions—because they physically group with the same genes across different genomes—and separate paralogs likely to have different functions—because they cluster with different groups of genes across genomes. Patterns of sequence conservation within and among paralog groups can then be used to identify “signature sequence motifs” that reliably identify the group of paralogs performing each functional role (Figure 1).

Tools to identify genomes encoding several genes of the same given family and extract the physical gene-neighborhood clusters associated with these genes are already available in the same integrative databases that already serve genome annotation pipelines such as IMG, MaGE or SEED, and we were able to use them to characterize the COG0720 family. Similarly, automated tools to identify signature motifs are now available. As shown in Figure 6, signature motifs identifying the three COG0720 families (PTPS-I, PTPS-II and PTPS-III) that we had created by manual analysis [55] could be recreated automatically using the Weblogo [172] and Web2logo platforms [173]. Once these signature motifs have been created, they can be used by tools such as HAMAP [168] to annotate all members of a multigene family, regardless of gene neighborhood (Figure 5). The value of using well-defined signature motifs as effective annotation tools has already been documented [174].

Based on our comparative genomics experience and approach, we propose the creation of a workflow, assembled from tools available, that we use and that are individually working well, to automate the identification of paralogs and their annotation. We believe that focusing on the difficult task of paralog annotation is needed, and will lead to valuable discoveries. In addition, improving the annotation process will be beneficial for the general research community.

## 8. Materials and Methods

### 8.1. Bioinformatic Analyses

The BLAST tools and resources at NCBI were routinely used [175]. Analysis of the distribution of genes of the COG0720 on the different kingdoms was performed using the tools of the IMG database (Phylogenetic Profiler for Single Genes) [176] and by a parsing of data obtained from the COG database [58].

Analysis of the phylogenetic distribution and physical clustering was performed in the SEED database [119] on the 11,411 genomes available at the time of the analysis (November 2015). Results are available in the “PTPS paralogs” subsystem on the public SEED server (http://pubseed.theseed.org//SubsysEditor.cgi?page=ShowSubsystem&subsystem=PTPS_paralogs) [119]. A subset of the analysis is summarized Figure 3. Clustal Ω [177] was used to verify the belonging of sequences to the PTPS-I, II, III or IV subfamilies, based on the strict presence of previously identified motifs [55] and re-annotated when needed. Sequences that were too short (typically with a motif partly missing or integrally missing) were re-annotated as “PTPS family” and flagged with the comment “#short, possible wrong start call.” We estimate that around 15% to 20% of the genes seem to be wrongly called. Sequences that align but for which no classification was possible based on the variability among the recognizable motif were annotated as “PTPS family.” The PATRIC annotations [178] were extracted and mapped to Uniprot genes when needed.

For sequence logo analysis and comparison, selected sequences corresponding to PTPS-I, II and III were extracted from the “PTPS paralogs subsystem” from genomes for which there is more than one PTPS (and are defined as paralogs) and for which there is evidence of functional coupling with genes previously identified involved in the specific PTPS pathway. Sequences were aligned with Clustal Ω [177]. Obtained alignments were trimmed with AliView-1.17.1 [179] to focus on the conserved motif area. The reference for the comparison contains all the extracted sequences and is used as a representative for the PTPS family. From the alignment, WebLogo [172] was used to create logos of the HGH motif region. Each logo was compared with the others in Two Sample Logo using the *t*-test option [173]. The sequences extracted are available in Appendix A.

### 8.2. Sequence Similarity Network (SSN) and Neighborhood

A script allowing the extraction of homologous sequences that are present in the same genome from the PTPS Exploration subsystem on PubSEED was used as input to generate a Sequence Similarity Network (SSN) and a Genome Neighborhood Network (GNN). Networks were also generated using PTPS sequences from the entire subsystem as input. The networks were generated using the EFI Enzyme Similarity Tool and Genome Neighborhood Tool [111,180] available at http://efi.igb.illinois.edu/efi-est/. The SSN was generated with an initial alignment score threshold of 15 using the custom Fasta option (option C). Cytoscape was used to visualize and edit the networks [181]. Uniprot accessions were imported into the SSNs and all nodes that did not contain a Uniprot accession were removed. The alignment score threshold of the networks was increased by deleting edges with increasing alignment score values in order to visualize separation of clusters. A table with manually determined annotations extracted from the “PTPS paralogs” subsystem was added to the network in order to test whether the clusters were iso-functional groups. The GNN was generated with default values using the SSN with only nodes containing a Uniprot accession at an alignment score threshold of 25 as input. This score was chosen based on the complete separation of PTPS families.

## 9. Conclusions

The issue of gene functional annotation quality will need to be solved for systems biology approaches to reach their full potential. As discussed here, paralogs are a major source of errors in gene functional annotation, and annotation quality cannot generally improve until paralog families are better annotated. Given recent advances in computational infrastructure and recent development of robust small-scale annotation efforts, we feel the time is right for large-scale annotation pipelines to begin integrating the methods used in small-scale expert-based annotations. With genome sequencing poised to generate a super-exponential increase in total sequence data over the coming years, failure to incorporate reliable gene-annotation methods now could leave us with a challenging “re-annotation” problem in the future. Although the methods described here are mainly applicable to prokaryotic genomes, as physical clustering is rare in eukaryotes, they may provide information informing annotation of the portion of eukaryote genomes that have bacterial homologs [48]. In addition to improving annotation quality, reliable identification of paralogs within multigene families is expected to provide identification of subfamilies with specific—but unknown—chemistry, potentially driving the discovery of previously unanticipated novel gene functions as shown with the examples in Table 1.

## Figures and Tables

**Figure 1 life-06-00039-f001:**
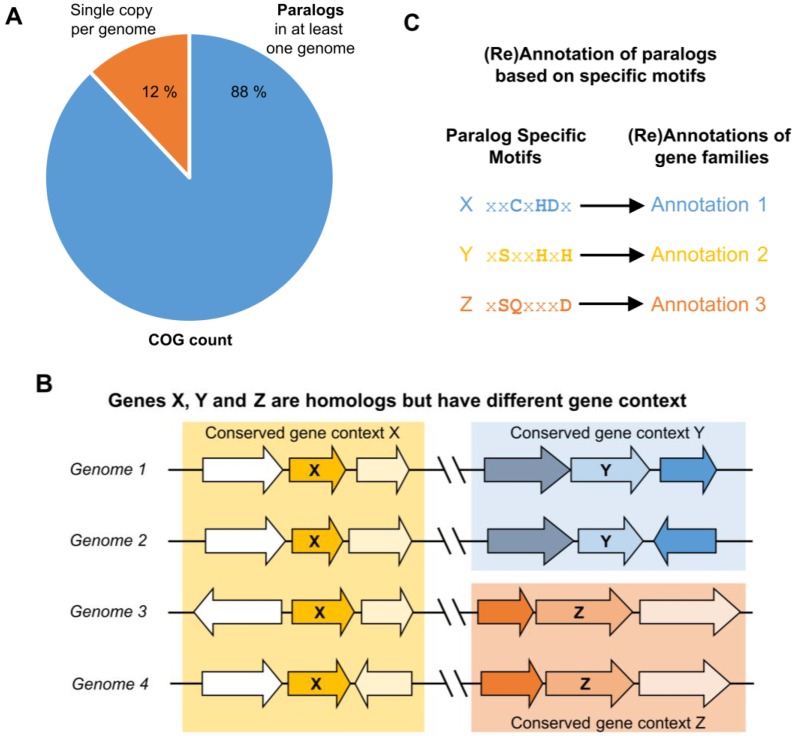
Differential paralog clustering can be used to identify non-isofunctional genes in multigene families. (**A**) Although single-copy genes can be accurately annotated by current high-throughput pipelines, they comprise a small percentage of the genome’s total gene count (12%, according to the National Center for Biotechnology Information (NCBI) Clusters of Orthologous Groups of proteins (COGs) database (2014 update) [58]). Genomes for which there is more than one homolog among a multigene family suggest the existence of paralogs among this family, and transfer of annotation based on homology is problematic among paralogs that can be non-isofunctional; (**B**) Paralogs with different flanking genes (genomic contexts) conserved across multiple species can be inferred as having different functions, even if their functions are not known; (**C**) Once paralogs are differentiated into provisional isofunctional groups by conserved genomic context, patterns of sequence conservation within and among isofunctional groups can be used to derive paralog-specific motifs, which can be used to annotate additional members of the isofunctional group for which there is no physical clustering conservation.

**Figure 2 life-06-00039-f002:**
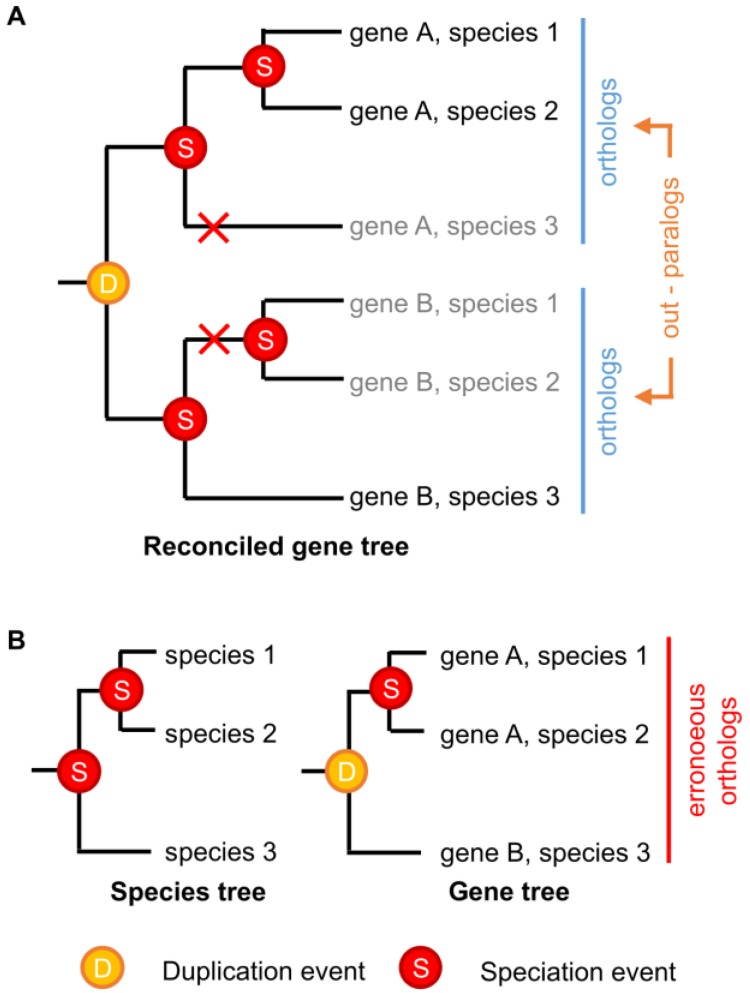
Differential gene loss can obscure the identification of paralogous genes. (**A**) A hypothetical example showing the historical evolutionary history of a simple multigene family across three species. Gene duplication and speciation events are indicated, with red crosses indicating gene losses. Grey names indicate unobservable “extinct” genes; (**B**) Even if the correct species tree were known with certainty, differential loss of paralogs across the species in (**A**) would result in the indicated gene tree, which is “correct,” given extant sequence data, but cannot observe the “extinct” genes. Gene-species tree reconciliation results in the erroneous grouping of paralogous genes A and B as orthologs.

**Figure 3 life-06-00039-f003:**
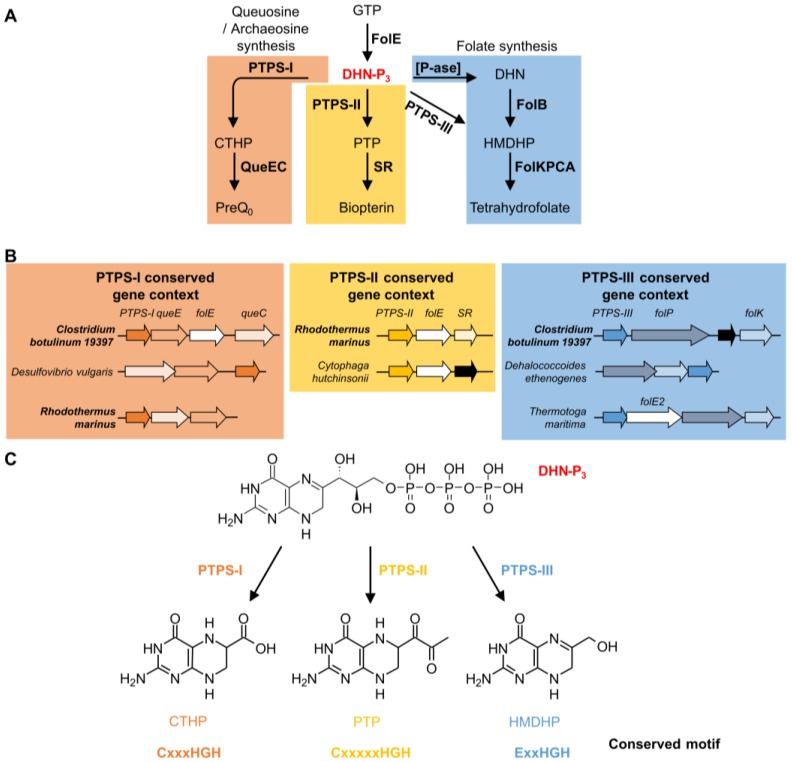
Functional roles of different PTPS subfamilies. Biosynthesis pathways in which PTPS-I, II and III are involved (**A**) and example of conserved gene context of PTPS-I, II and III linking them with these pathways (**B**); Note that *Clostridium botulinum 19397* and *Rhodothermus marinus* are among the organisms having more than one PTPS gene belonging to different conserved gene contexts. Specific reactions catalyzed by PTPS-I, II and III, and conserved motifs identified (**C**). Abbreviations: GTP: guanosine triphosphate; FolE: GTP cyclohydrolase I; FolE2: GTP cyclohydrolase II; DHN-P3: dihydroneopterin-triphosphate; CTHP: 6-Carboxytetrahydropterin; PTP: 6-pyruvoyl-tetrahydropterin; HMDHP: 6-hydroxymethyldihydropterin; PreQ0: 7-cyano-7-deazaguanosine; QueE: 7-carboxy-7-deazaguanine synthase; QueC: 7-cyano-7-deazaguanine synthase; SR: sepiapterin reductase; DHN: dihydroneopterin; [P-ase]: phosphatase; FolB: Dihydroneopterin aldolase; FolK: 6-Hydroxymethyl-7,8-dihydropterin pyrophosphokinase; FolP: Dihydropteroate synthase; FolC: Dihydrofolate:folylpolyglutamate synthase; FolA: Dihydrofolate reductase.

**Figure 4 life-06-00039-f004:**
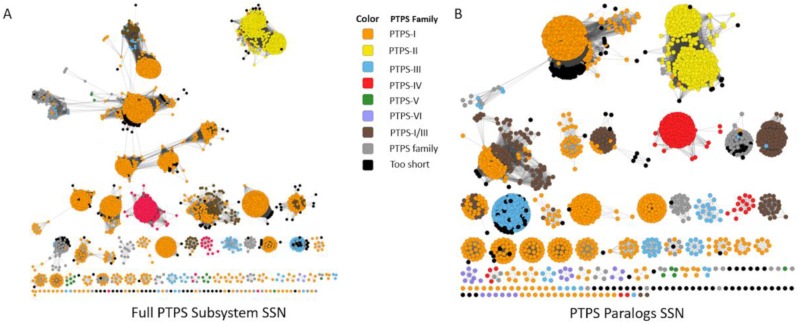
Sequence Similarity Networks Generated from the “PTPS paralog” Subsystem. An SSN was generated from the sequences extracted from the “PTPS paralog” subsystem, at alignment score threshold of 30 (**A**); A specific subset of sequences, limited to only to PTPS homologs in the same genomes (paralogs), were used to generate an SSN with identical parameters (**B**).

**Figure 5 life-06-00039-f005:**
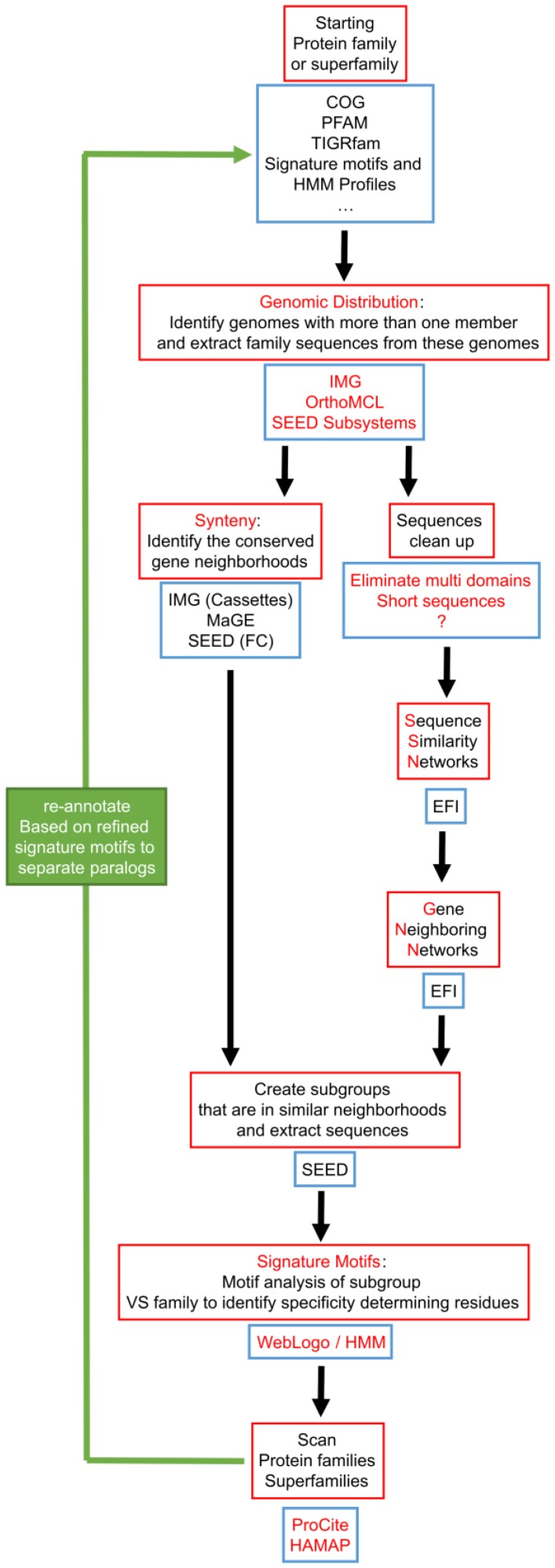
Proposed workflow for improving paralog annotation that integrates existing tools. The key features of this workflow are: focusing on genomes that already have paralogs, and building signature motifs only on groups of sequences that share similar physical neighborhoods. Red boxes describe the different required operations, and blue boxes list existing databases or software that provide the tools to perform these operations. The use of SSN and GNN is proposed as a strategy to identify the groups of sequences to build signatures as we feel their pipeline could become more automatable once it has matured. FC: Functionally coupled.

**Figure 6 life-06-00039-f006:**
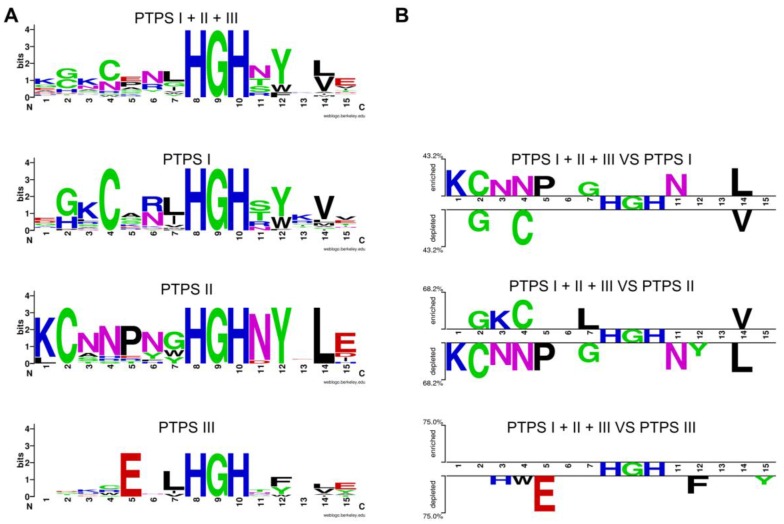
Motifs identified in paralogs from the same genome based on comparing with homologs with conserved physical clustering. Logos of the region containing the signature sequence (amino acids 23 to 38 in *E. coli* Uniprot P65870) from different COG0720 families (**A**); The choice of the sequences for the logos was as followed: starting from genomes that had two or more members of the family, only sequences that were in similar gene neighborhoods were chosen to generate the logos. The set of sequences used is given in Appendix A. The logos were generated using Weblogo [172]. Logos from the different subfamilies were compared using the Two Sample Logo platform [173] (**B**). This automated comparison tool objectively identifies key residues for each subfamily.

**Table 1 life-06-00039-t001:** Examples of paralogs with different functions elucidated by comparative genomic approaches.

COG ^1^	Paralog Subfamilies			Ref.
0780 and 0302	QueF (Queuosine synthesis)	FolE (Tetrahydrofolate synthesis)	QueF-Like (Aracheosine synthesis)	[45,46]
1539	FolX (Tetrahydromonapterin synthesis)	FolB (Tetrahydrofolate synthesis)		[47]
1028 and 0262	FolM (Tetrahydromonapterin synthesis)	FadG (fatty acid synthesis)	FolA (Tetrahydrofolate synthesis)	[47]
0009	YciO (Unknown function)	YrdC/TsaC (t^6^A synthesis)		[48]
1509	YjeK (Protein modification)	LamB (Lysine degradation)		[49]
2269 and 1190	YjeA (Protein modification)	LysRS (Protein synthesis)		[49]
0354 and 0404	YgfZ (Iron-sulfur cluster repair)	GcvT (One carbon metabolism)		[50]
2102	Dph6 (Diphtamide synthesis)	DUF71-B12 group (function unknown, B12 salvage)		[51]
5424	PqqC (PQQ synthesis)	CT610 (Para-aminobenzoate synthesis)		[52]
1478	CofE (F420 synthesis)	CT611 (Tetrahydrofolate synthesis)		[53]
1901	TrmY Archaeal m1Psi54 methylase	Bacterial unknown methylase		[54]
0720	PTPS-I (QueD, Queuosine synthesis)	PTPS-II (Biopterin synthesis)	PTPS-III (Folate synthesis)	[55]
0523	15 subfamilies identified			[56]
0212	5-formyltetrahydrofolate cycloligase (5-FCL)	Thiamin metabolism		[57]

^1^ Data obtained from the Clusters of Orthologous Groups of proteins (COGs) database, 2003 COGs, 2014 update [58]. Please note that if some paralog pairs are currently mapping to more than one COG, it was not the case at the time of the reported studies.

**Table 2 life-06-00039-t002:** COG0720 genome count from diverse genomes ^1^.

Domain	Genome Count with COG0720	COG0720 Gene Count	Genomes with COG0720 Paralogs	Total Genomes
Archaea	561	698	134	771
Bacteria	636	821	164	1056
Eukaryota	60	65	5	220

^1^ Data extracted from IMG using the search function as of 30 November 2015. Genome list is provided as Appendix A.

**Table 3 life-06-00039-t003:** Annotations and information available in diverse databases for the genes belonging to the classes PTPS-I and PTPS-III of *Pyrococcus furiosus* DSM 3638 ^1^.

Annotation	Database Identifier	Annotation	Database Identifier	Annotation
NCBI annotation	WP_011011332.1	**6-carboxy-5,6,7,8-tetrahydropterin synthase (NCBI Reference Sequence)**	WP_011012422.1	6-pyruvoyltetrahydropterin synthase (NCBI Reference Sequence)
Ensemble bacteria	AAL80343	putative 6-pyruvoyl tetrahydrobiopterin synthase (PF0219)	AAL81402	hypothetical protein (PF1278)
PATRIC (uses the RAST annotation pipeline)	fig186497.12.peg.227	**Queuosine biosynthesis QueD, PTPS-I**	fig186497.12.peg.1340	**Folate biosynthesis protein PTPS-III, catalyzes a reaction that bypasses dihydroneopterin aldolase (FolB)**
		putative 6-pyruvoyl tetrahydrobiopterin synthase		hypothetical protein
MaGe	PF0219	putative 6-pyruvoyl tetrahydrobiopterin synthase automatic/finished	PF1278	hypothetical protein automatic/finished
		NCBI RefSeq Annotation: putative 6-pyruvoyl tetrahydrobiopterin synthase		NCBI RefSeq Annotation: hypothetical protein
		TrEMBL annotation: Putative 6-pyruvoyl tetrahydrobiopterin synthase		TrEMBL annotation: Dihydroneopterin monophosphate aldolase
IMG	638172701 ^2^	preQ(0) biosynthesis protein QueD	638173858 ^2^	hypothetical protein
	2625830234 ^3^	6-pyruvoyltetrahydropterin/**6-carboxytetrahydropterin synthase**	2625831353 ^3^	6-pyruvoyltetrahydropterin/6-carboxytetrahydropterin synthase

^1^ Annotations considered correct are highlighted in bold; ^2^ Initial annotation; ^3^ Re-annotated genome.

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
