# Peer review of "Functional Annotations of Paralogs: A Blessing and a Curse"

_life, 2016, doi:10.3390/life6030039_

Round 1

Reviewer 1 Report

This article is a valuable review of approaches for functional annotation of relatives in protein families. Many of the major resources are described in the article. I assume that this manuscript is being considered as a review rather than a research paper, as there is no substantial assessment of the different resources just an analysis using one example family.

In fact, I found the assessment of the platforms providing functional annotations of proteins rather strange. Many of these resources are not described as paralog identification resources by their authors. They have mostly been established to suggest functional properties for query proteins. It is not very helpful to make judgements of these resources on the basis of a single example tested. It would help if this was presented in a more constructive way and these caveats made clear.

Furthermore, whilst the review is valuable in pointing out the different approaches that could be used to improve ortholog/paralog distinction and function annotation, the implementation of most of these approaches is very challenging and combining these approaches would also be non-trivial. A certain level of error would be inevitable.

The manual strategy used by the authors to separate COG0720 was not very clear to me from the text and figures nor how it would easily translate into an automated pipeline. This should be made much clearer. Their schematic of the method shown in figure 1 is very idealised and the figure for their PTPS example doesn’t show the same simplicity and would be quite challenging for an algorithmic approach.

To be honest, I found the review somewhat naïve. I suspect that the workflow illustrated in figure 5 is similar to workflows already in use by large teams of curators in Swiss-Prot and HAMAP who use these tools routinely – but separately. Each tool requires considerable resources to build. To build a computational pipeline that somehow automatically combines all these steps is very ambitious (and risky) and if it was feasible I suspect Swiss-Prot would already have done it!

Furthermore, different families can behave differently and what works well for one is not necessarily the best approach for another. So I’m uncertain that any single pipeline will ever be completely reliable for every family, hence the importance of manual inspection and curation for very detailed analyses of protein families.

However, the review is still valuable in highlighting tools that can be beneficial when used together (ie by a curator consulting the predictions from each tool) and in reinforcing the need for funding to support manual curations effects such as HAMAP!

Minor points

1. Table 1 is confusing – why are some paralog pairs mapping to more than one COG cluster. It would be helpful to know the sequence identity for these example pairs. Its not clear to me what makes these examples particularly interesting. I’m assuming they were difficult to identify by standard sequence comparison based techniques but that’s clear from the table.

2. What version of COG is used? – this resource is very out of date. It would have been better to refer to EggNog.

3. To what extent does similarity in genome context imply similarity in function? Have any quantitative analyses been performed?  Surely there is the same scope for divergence in function as with considering divergence in proteins having only one residue mutation. In fact the authors show that Genetic Neighbourhood Networks don’t really work that well either!

4. Relatively recent work by the group of Christophe Dessimov has also suggested that Orthologs are not always as functionally conserved as paralogs. The authors might want to cite this work.

5. The manuscript seems rather hastily written in parts (some resources were not given their correct names) and some paragraphs were very confusing. I am not aware of SFLD using fully automatic methods for generating functional subfamilies. They use experimental data and manual curation although their classification is guided by sequence similarity networks (SSNs). The manual inspection and intervention combined with the experimental analyses is what makes SFLD such a reliable resource for benchmarking automated approaches. CATH-Gene3D FunFams (there is no resource called CATH-3D) comprises 110,000 functional families (Das et al, Bioinformatics, ref). CATH-Gene3D FunFams is applied on a whole genome scale ie all proteins from completed genomes that can be assigned to CATH-Gene3D superfamilies are functionally classified.  

6. It would be good to indicate in the abstract that the article focuses on only one multi-gene/ multi-functional family - the COG0720 family

7.  The article has a very detailed background of the field which comprises many sections even after the Introduction that makes the research article appear more like a review. Also the manuscript would benefit from having a clearly defined Results section.

8. CATH-3D should be corrected to CATH-Gene3D resource throughout the manuscript (also in page 11) and the number of families needs to be updated as the resource currently provides >2730 superfamilies classified into 110,000 families. 

9. In the first paragraph of the fourth section (The COG0720 case study), please mention whether default BLAST parameters were used in the BLAST search otherwise the BLAST parameters should be mentioned. 

10. In the second paragraph of the fourth section (The COG0720 case study), please mention the full form of IMG

11. In the third paragraph of the fourth section (The COG0720 case study), it would be useful to mention or list the "different types of comparative genomics evidence" used to identify the COG0720 sub-families. This article mentions only three PTPS subfamilies, however, the Phillips et. al., 2011 paper cited for this work had mentioned four PTPS subfamilies and up to six PTPS motifs in the original article. Please explain any reasons for not considering the other COG0720 subfamilies in this analysis. 

12. In the second paragraph of the fifth section, Please correct the sentence: "This is very useful to identify fusion proteins that are a well know misannotation problems."

13. Please capitalise Patric in Table 3 so that is consistent throughout the manuscript.

14. In Figure 5, in the fourth last block of the flowchart titled Signature Motifs, please specify what are determining residues - specificity determining residues? 

15. It would also be good to provide citations for WebLogo and Two Sample Logo in the caption of Figure 6. 

16. In the Methods section, it would be useful to list the name of the tools used in the IMG/COG/SEED databases for the different analyses.

Author Response

This article is a valuable review of approaches for functional annotation of relatives in protein families. Many of the major resources are described in the article. I assume that this manuscript is being considered as a review rather than a research paper, as there is no substantial assessment of the different resources just an analysis using one example family.

This paper is not in the classical review/original research format but is purposely a blend. This format had been discussed with the Life editors prior to submission and the second reviewer summarizes well the reasoning behind this format as quoted here: “This emphasizes the anticipated broad impact of the current manuscript, which combines elements of a visionary topic review, actual research article and hands-on tutorial. In fact, the latter is probably the most accurate classification of this MS format, which, by itself, is very contemporary and goes beyond a pregenomic classification (research paper vs review).”  

In fact, I found the assessment of the platforms providing functional annotations of proteins rather strange. Many of these resources are not described as paralog identification resources by their authors. They have mostly been established to suggest functional properties for query proteins. It is not very helpful to make judgements of these resources on the basis of a single example tested. It would help if this was presented in a more constructive way and these caveats made clear.

We respectfully disagree with the reviewer. The results of the genome annotation process are presented in databases that integrate metadata from others sources to suggest functional properties for query proteins. The difficulty and problem of the annotation of multigene families is known, as several of the annotation platforms state (in the manuscript or documentation associated) that they possess tools to explore proteins families, or to separate paralogs.

(See below for extracted documentation from manuscripts or online documentation.)

However, when one delves in the nitty-gritty details of the annotation process, it is clear that paralogs are not separated in the standard annotation process.

We chose one complex example that we know well - the COG0720 family - to illustrate this point. This is mainly to make biologists that are not well versed in the annotation process aware of the problem. In general annotation of paralog families is so poor that a comprehensive study comparing annotation pipelines with recall and precision statistics would be totally impossible to do, hence our focus on one specific family.

NCBI annotation pipeline

http://www.ncbi.nlm.nih.gov/books/NBK174280/

Prokaryotic Genome Annotation Pipeline

Tatusova, et al, 2013

“Protein naming

The final component of the pipeline is identifying protein function and naming the protein product of the coding region. Assignment of a predicted model to a cluster for purposes of naming is based on protein homology to members of the cluster: we require high coverage, high-scoring alignments to at least three members of the same cluster in order to assign a protein to a cluster.”

Annotation of gene based on the belonging into a cluster. Clusters are defined as: Conserved proteins from curated clusters, Clusters of Orthologous Groups and NCBI Prokaryotic Clusters.

Annotation, is thus based on clusters of orthologous proteins. We could expect paralogous proteins to belong to different clusters, and be annotated differently.

Ensembl genomes

Documentation available on their website.

http://ensemblgenomes.org/info/data/peptide_compara

“Peptide comparative genomics

Comparative analysis at the peptide level is performed for each Ensembl Genomes division, and additionally a pan-taxonomic comparative analysis is performed for a separate set of representative species from across the taxonomic space. The division-specific comparative analysis generate an implied evolutionary history for every gene family, and which can be visualised either as a "gene tree", or in the form of derived lists of orthologues and paralogues. The detailed documentation of the analysis done can be found here.”

RAST

RASTtk: A modular and extensible implementation of the RAST algorithm for building custom annotation pipelines and annotating batches of genomes

Brettin et al., 2015, Scientific reports

“Annotating proteins with k-mers. Historically, RAST has made heavy use of the FIGfam collection maintained within the SEED project6.

FIGfams are protein families in which it is believed that all members of the same family share an identical function and were derived from a common ancestor (i.e., they are all isofunctional homologs). The original implementation of the k-mer-based assignment of function was based on the use of signature k-mers3,19

.A signature k-mer is

defined as an 8-mer amino acid sequence in which the vast majority (over 80%) of occurrences are found within FIGfams sharing a com- mon function, and that do not occur in any FIGfam with a different function. For example, a k-mer for which 93% of the occurrences within the FIGfam collection were in families implementing the function SSU ribosomal protein S13p (S18e) would be considered a ‘‘signature of function’’. In this case, the signatures depend critically on the FIGfams.”

MAGE annotation pipeline

MaGe: a microbial genome annotation system supported by synteny results

Vallenet et al., 2006, NAR

“For assigning function to novel proteins, gene context approaches can complement the classical homology-based gene annotation. These ‘nonhomology-based’ inference methods rely on the fact that functionally associated proteins are encoded by genes that share similar selection pressures. In most of the proposed methods (14,37,38), orthologous pairs of proteins satisfy the bi-directional best hit (BBH) criterion, based on blast and/or Smith–Waterman (39) comparisons of complete genomes with one another. An innovative aspect of our approach is that we offer the possibility of retaining more than one homologous gene. Pairwise comparisons between predicted protein sequences of the studied genome and the proteins of another genome allow computation of ranked hits and BBH (for each protein, the three best hits are kept). Putative orthologous relations between two genomes are defined as gene couples satisfying the BBH criterion or an alignment threshold (generally, a minimum of 30% sequence identity on 80% of the length of the smallest protein). These relations are subsequently used to search for conserved gene clusters, e.g. synteny groups among several bacterial genomes. Our method, called the Syntonizer, is based on an exact graph-theoretical approach (40). This method allows for multiple correspondences between genes and, thus, paralogy relations and/or gene fusions are easily detected.

Clear mention of the capacity of identifying orthologs and paralogs.

IMG annotation pipeline

Paper “Improving Microbial Genome Annotations in an Integrated Database Context”, Chen et al., 2013, Plos ONE.

“In cases of protein families with a large number of paralogs and frequent events of gene duplication and gene loss, the discrepancies between gene sets assigned to the same functional role by different annotation resources are even more pronounced. Sometimes they are due to annotation errors, but quite often they are indicative of the presence of a subfamily with different substrate specificity (or otherwise different function), which has not been identified as such by any of the annotation resources.

IMG provides tools for further exploration of such subfamilies via the analysis of the chromosomal synteny and phylogenetic occurrence profiles [18,19]. An illustration of IMG tools developed for assessment of the consistency of functional annotation is provided below, using KO terms as an example.”

Even if there are tools described as existing in the databases, the annotations are however not actually changed based on the results of these tools analysis. The proposed pipeline would rely on the tools available in these databases.

Furthermore, whilst the review is valuable in pointing out the different approaches that could be used to improve ortholog/paralog distinction and function annotation, the implementation of most of these approaches is very challenging and combining these approaches would also be non-trivial. A certain level of error would be inevitable.

The manual strategy used by the authors to separate COG0720 was not very clear to me from the text and figures nor how it would easily translate into an automated pipeline. This should be made much clearer. Their schematic of the method shown in figure 1 is very idealised and the figure for their PTPS example doesn’t show the same simplicity and would be quite challenging for an algorithmic approach.

To be honest, I found the review somewhat naïve. I suspect that the workflow illustrated in figure 5 is similar to workflows already in use by large teams of curators in Swiss-Prot and HAMAP who use these tools routinely – but separately. Each tool requires considerable resources to build. To build a computational pipeline that somehow automatically combines all these steps is very ambitious (and risky) and if it was feasible I suspect Swiss-Prot would already have done it!

Furthermore, different families can behave differently and what works well for one is not necessarily the best approach for another. So I’m uncertain that any single pipeline will ever be completely reliable for every family, hence the importance of manual inspection and curation for very detailed analyses of protein families.

However, the review is still valuable in highlighting tools that can be beneficial when used together (ie by a curator consulting the predictions from each tool) and in reinforcing the need for funding to support manual curations effects such as HAMAP!

The main idea for the pipeline proposed in the paper is to start with genomes that contain paralogs as the nucleus to differentiate the subfamilies. We thought Figure 1 made this point clearly but we decided to edit the figure and associated legend to emphasize this better. This is the strategy that we found to be the most efficient to manually separate paralog subfamilies as shown with all the examples in Table 1 and in the COG0720 case study section. We edited Figure 3 to reflect that it is the same global strategy presented in Figure 1 with the specific COG0720 example. Because the strategy we use reposes on tools that are available in most of the databases we consult, we think an automation may be possible.

It might sound a “naïve” idea but none of the existing platforms actually use this simple but effective strategy even if some tools to that are based on the same principle have been developed. Our message here is that by starting the analysis in this simple fashion automation might become possible. As correctly stated by the reviewer all the pieces of the pipeline are already in place. Combining them in a pipeline is far from easy or as the reviewer said “swissprot would already have done it”, but what we are proposing here is that by starting first on genomes with paralogs, a pipeline might become feasible. This pipeline might not be fully automatable but could greatly reduce the manual work needed to correctly annotate a paralog family.

Minor points

Table 1 is confusing – why are some paralog pairs mapping to more than one COG cluster. It would be helpful to know the sequence identity for these example pairs. Its not clear to me what makes these examples particularly interesting. I’m assuming they were difficult to identify by standard sequence comparison based techniques but that’s clear from the table.

None of these examples had been separated in functional subfamilies initially, using standard sequence comparison techniques. If some paralog pairs are now mapping to more than one COG, it was however not the case at the time of the reported studies. Characterization was then taken into account to subdivide the original COGs in different groups. A reflecting note has been added as a comment of the table.

2. What version of COG is used? – this resource is very out of date. It would have been better to refer to EggNog.

The version of the COG database used is the last version, “2003 COGs, 2014 update”, (http://www.ncbi.nlm.nih.gov/COG/), with the associated reference Nucleic Acids Res. 2015 Jan;43:D261-D269. For clarity, the version has been added to the Table 1, in the legend of Figure 1, and clarified in lines the manuscript.

3. To what extent does similarity in genome context imply similarity in function? Have any quantitative analyses been performed?  Surely there is the same scope for divergence in function as with considering divergence in proteins having only one residue mutation. In fact the authors show that Genetic Neighbourhood Networks don’t really work that well either!

Actually there is very strong evidence that similarity in genome context is a strong indication of similarity of function (Using genome-context data to identify specific types of functional associations in pathway/genome databases, Green and Karp, 2007, Bioinformatics; Conservation of gene order: a fingerprint of proteins that physically interact, Dandekar et al., 1998, TIBS; Overbeek et al., 1998; Computational method to assign microbial genes to pathways, Pellegrini et al., 2001, Journal of Cellular Biochemistry; The society of genes: networks of functional links between genes from comparative genomics, Yanai et al., 2002, Genomes Biology). A paragraph reintroducing the genome context approach has been added to the manuscript.

We did not state that the GENOME NEIGHBORHOOD NETWORKS do not work well. The information provided by GNN is actually very similar to a comparative genomics approach. However, the SSN creation and the use of the data generated to create GNN is not simple and straightforward, it requires a huge activation barrier. We would like to state that useful information is presented by GNT tools, but we feel it is not easily accessible.

4. Relatively recent work by the group of Christophe Dessimov has also suggested that Orthologs are not always as functionally conserved as paralogs. The authors might want to cite this work.

The work of C. Dessimoz is already cited in the paper. See reference:

Altenhoff, A. M.; Studer, R. A.; Robinson-Rechavi, M.; Dessimoz, C. Resolving the ortholog conjecture: Orthologs tend to be weakly, but significantly, more similar in function than paralogs. PLoS Comput. Biol. 2012, 8, e1002514.

The interpretation of the reviewer seems to be reversed to what is stated in that paper.

5. The manuscript seems rather hastily written in parts (some resources were not given their correct names) and some paragraphs were very confusing. I am not aware of SFLD using fully automatic methods for generating functional subfamilies. They use experimental data and manual curation although their classification is guided by sequence similarity networks (SSNs). The manual inspection and intervention combined with the experimental analyses is what makes SFLD such a reliable resource for benchmarking automated approaches. CATH-Gene3D FunFams (there is no resource called CATH-3D) comprises 110,000 functional families (Das et al, Bioinformatics, ref). CATH-Gene3D FunFams is applied on a whole genome scale ie all proteins from completed genomes that can be assigned to CATH-Gene3D superfamilies are functionally classified.  

We edited the manuscript were we think clarifications were needed. Deletions, corrections and precisions were made.

6. It would be good to indicate in the abstract that the article focuses on only one multi-gene/ multi-functional family - the COG0720 family

The abstract was edited, as suggested.

7.  The article has a very detailed background of the field which comprises many sections even after the Introduction that makes the research article appear more like a review. Also the manuscript would benefit from having a clearly defined Results section.

The format of the manuscript was discussed with the editor prior to submission.

8. CATH-3D should be corrected to CATH-Gene3D resource throughout the manuscript (also in page 11) and the number of families needs to be updated as the resource currently provides >2730 superfamilies classified into 110,000 families. 

The relevant section was edited, as suggested

9. In the first paragraph of the fourth section (The COG0720 case study), please mention whether default BLAST parameters were used in the BLAST search otherwise the BLAST parameters should be mentioned. 

The corresponding paragraph was edited as suggested.

10. In the second paragraph of the fourth section (The COG0720 case study), please mention the full form of IMG

The corresponding paragraph was edited as suggested.

11. In the third paragraph of the fourth section (The COG0720 case study), it would be useful to mention or list the "different types of comparative genomics evidence" used to identify the COG0720 sub-families. This article mentions only three PTPS subfamilies, however, the Phillips et. al., 2011 paper cited for this work had mentioned four PTPS subfamilies and up to six PTPS motifs in the original article. Please explain any reasons for not considering the other COG0720 subfamilies in this analysis. 

The manuscript was edited to precise the comparative genomics approach.

The manuscript initially mentioned the presence of 6 PTPS subfamilies, so no changes were made. This information is also clearly present in the Figure 4.

12. In the second paragraph of the fifth section, Please correct the sentence: "This is very useful to identify fusion proteins that are a well know misannotation problems."

The sentence has been corrected.

13. Please capitalise Patric in Table 3 so that is consistent throughout the manuscript.

Changes were made, as requested.

14. In Figure 5, in the fourth last block of the flowchart titled Signature Motifs, please specify what are determining residues - specificity determining residues? 

Changes were made, as requested.

15. It would also be good to provide citations for WebLogo and Two Sample Logo in the caption of Figure 6. 

Citation were added in the Figures, as suggested.

16. In the Methods section, it would be useful to list the name of the tools used in the IMG/COG/SEED databases for the different analyses.

Precisions were made, as requested.

Reviewer 2 Report

The manuscript by Zallot et al. provides an insightful analysis of a highly significant problem of accurate and confident assignment of functions to paralogous (homologous but non-orthologous) proteins/genes. This question is tackled from both crucial perspectives, as a general problem of genomic annotation pipelines (bioinformatics problem), as well as a specific task of an individual database user trying to figure out a function of her favorite gene (biological problem). For this and other reasons mentioned below, this article is of broad interest, as its audience would include both, professional genome annotators/tool developers and, more importantly, users-biologists that still tend to fall into one of the two major extremes: (i) those who blindly trust database annotations, and (ii) those who distrust them altogether considering any gene function “unknown, until experimentally tested”. Both extremes are unfortunate, and it is hard to tell which one is more counterproductive in the postgenomic era. Remarkably and absolutely correctly, the authors point our that the paralog challenge may be, in fact, turned into an opportunity to expand functional knowledge of functionally heterogeneous protein families. From the individual user perspective, it is equivalent to a notion that any homology-based link, even to proteins with distinct functions, is useful to establish the function of “my protein of interest”. Of course to do it right, one has to recognize that: (a) “my protein” is a paralog rather than ortholog of those with well-established functions, and, hence the current annotation of it is a suspect; and (b) other types of comparative genomics-derived evidence, beyond homology (often referred to as “genomic context”), should be explored to arrive to its functional assignment (which, if made for the first time, is equivalent to discovery, a worthy subject of experimental testing). While the appreciation of the problem and a concept of integrating various types of genomic evidence for accurate functional assignment of paralogs emerged in the early days of genomics, the public awareness of both aspects is still very limited. This emphasizes the anticipated broad impact of the current manuscript, which combines elements of a visionary topic review, actual research article and hands-on tutorial. In fact, the latter is probably the most accurate classification of this MS format, which, by itself, is very contemporary and goes beyond a pregenomic classification (research paper vs review). The corresponding author of this article is among the pioneers of the genomic context approach to gene function discovery, whose work helped its maturation and validation via multiple case studies. Overall this MS is highly recommended for publication “as is”, with minor (and optional) edits (As in a couple of examples below):

FIGURE 1/Legend:  Of the three hypothetical genes (X, Y and Z), the third one is shown as having a conext, which is not marked as “conserved”, but in the legend, the authors talk more generally about conservation of the context “across multiple species” (which is appropriate). This apparent discrepancy may lead to confusion, and it ay be easily avoided via minor edits

More generally: I feel that the paper would benefit from a brief discussion/comments that a notion of a conserved “physical clustering” of functionally linked (coupled, involved in the same biological process) genes in microbes is historically and conceptually related (albeit distinct) to a notion of “coregulation” via operons and regulons. While obvious to authors, it may help a broader audience to appreciate the fundamentals of the genome context-based approach to gene function assignment and discovery.

Author Response

The manuscript by Zallot et al. provides an insightful analysis of a highly significant problem of accurate and confident assignment of functions to paralogous (homologous but non-orthologous) proteins/genes. This question is tackled from both crucial perspectives, as a general problem of genomic annotation pipelines (bioinformatics problem), as well as a specific task of an individual database user trying to figure out a function of her favorite gene (biological problem). For this and other reasons mentioned below, this article is of broad interest, as its audience would include both, professional genome annotators/tool developers and, more importantly, users-biologists that still tend to fall into one of the two major extremes: (i) those who blindly trust database annotations, and (ii) those who distrust them altogether considering any gene function “unknown, until experimentally tested”. Both extremes are unfortunate, and it is hard to tell which one is more counterproductive in the postgenomic era. Remarkably and absolutely correctly, the authors point our that the paralog challenge may be, in fact, turned into an opportunity to expand functional knowledge of functionally heterogeneous protein families. From the individual user perspective, it is equivalent to a notion that any homology-based link, even to proteins with distinct functions, is useful to establish the function of “my protein of interest”. Of course to do it right, one has to recognize that: (a) “my protein” is a paralog rather than ortholog of those with well-established functions, and, hence the current annotation of it is a suspect; and (b) other types of comparative genomics-derived evidence, beyond homology (often referred to as “genomic context”), should be explored to arrive to its functional assignment (which, if made for the first time, is equivalent to discovery, a worthy subject of experimental testing). While the appreciation of the problem and a concept of integrating various types of genomic evidence for accurate functional assignment of paralogs emerged in the early days of genomics, the public awareness of both aspects is still very limited. This emphasizes the anticipated broad impact of the current manuscript, which combines elements of a visionary topic review, actual research article and hands-on tutorial. In fact, the latter is probably the most accurate classification of this MS format, which, by itself, is very contemporary and goes beyond a pregenomic classification (research paper vs review). The corresponding author of this article is among the pioneers of the genomic context approach to gene function discovery, whose work helped its maturation and validation via multiple case studies. Overall this MS is highly recommended for publication “as is”, with minor (and optional) edits (As in a couple of examples below):

FIGURE 1/Legend:  Of the three hypothetical genes (X, Y and Z), the third one is shown as having a conext, which is not marked as “conserved”, but in the legend, the authors talk more generally about conservation of the context “across multiple species” (which is appropriate). This apparent discrepancy may lead to confusion, and it ay be easily avoided via minor edits

We modified the Figure 1, (also as requested by reviewer 3), and also integrated the mention “Conserved gene context Z”, with the addition of a fourth genome. We hope the information is presented more clearly.

More generally: I feel that the paper would benefit from a brief discussion/comments that a notion of a conserved “physical clustering” of functionally linked (coupled, involved in the same biological process) genes in microbes is historically and conceptually related (albeit distinct) to a notion of “coregulation” via operons and regulons. While obvious to authors, it may help a broader audience to appreciate the fundamentals of the genome context-based approach to gene function assignment and discovery.

A brief paragraph was added regarding the physical clustering of genes and how it has been proposed to be helpful for identifying associations between genes, and how co-regulation could have been a motor in such linking.

Reviewer 3 Report

This paper addressed the question of how to correctly annotate gene function, and focused on the functional annotation of duplicate genes. They first pointed out that the functional annotation based on sequence similarity to transfer gene function annotation are unwarranted. Then they described the caveats in ortholog/paralog identification. By analyzing the latest COGs, they suggested that sequence-similarity based approaches might be error-prone in orthology/paralogy inference. Furthermore, they showed that comparative-genomic information, such as physical clustering, can be used to separate and annotate paralogs, and suggested that combination of phylogenetic/sequence-similarity information with comparative-genomics information could largely improve the reliability of separating functional groups.  Next, they showed that the COG0720 family has at least three subfamilies, and used it as an example for the later analysis. They examined several genome annotation pipelines to annotate two proteins of COG0720, which belong to the different subfamily, and found most pipelines could not separate the two paralogs correctly. Lastly, they proposed a workflow to annotate the function of paralogs by integrating evolutionary, functional information and comparative genomics with existing tools.

Overall, this is a interesting paper, which addressed an important question in functional annotation of genes. However, the whole structure of the manuscript is very loose and should be largely revised to be more organized and more concise. 

The specific comments are:

1) The three sections: protein function and evolution, identifying orthologs and paralogs in practice, lessons from comparative genomics, should be merged into the introduction section or later discussion section, and should be more concisely written. 

2) The three sections: the COG0720 case study,  current methods for paralog annotation in genome annotation pipelines, and integration of tools for paralog separation in a workflow, should be merged into one result section, and be more concisely described.

3) When some abbreviations were mentioned at the first time, they did not follow the full name of the phrase, for example "COGs" first appeared in the Table 1, but until page 6, the authors mentioned "Clusters of Orhologous Groups (COGs)". Furthermore, the "Orhologous" should be "Orthologous". Another example, line 100, the EC number, what does "EC" stand for?

4) Figure 1A might be better shown as a pie plot.

5) Page 14, the S6 Text can't be found.

Author Response

This paper addressed the question of how to correctly annotate gene function, and focused on the functional annotation of duplicate genes. They first pointed out that the functional annotation based on sequence similarity to transfer gene function annotation are unwarranted. Then they described the caveats in ortholog/paralog identification. By analyzing the latest COGs, they suggested that sequence-similarity based approaches might be error-prone in orthology/paralogy inference. Furthermore, they showed that comparative-genomic information, such as physical clustering, can be used to separate and annotate paralogs, and suggested that combination of phylogenetic/sequence-similarity information with comparative-genomics information could largely improve the reliability of separating functional groups.  Next, they showed that the COG0720 family has at least three subfamilies, and used it as an example for the later analysis. They examined several genome annotation pipelines to annotate two proteins of COG0720, which belong to the different subfamily, and found most pipelines could not separate the two paralogs correctly. Lastly, they proposed a workflow to annotate the function of paralogs by integrating evolutionary, functional information and comparative genomics with existing tools.

Overall, this is a interesting paper, which addressed an important question in functional annotation of genes. However, the whole structure of the manuscript is very loose and should be largely revised to be more organized and more concise. 

The specific comments are:

1) The three sections: protein function and evolution, identifying orthologs and paralogs in practice, lessons from comparative genomics, should be merged into the introduction section or later discussion section, and should be more concisely written. 

2) The three sections: the COG0720 case study,  current methods for paralog annotation in genome annotation pipelines, and integration of tools for paralog separation in a workflow, should be merged into one result section, and be more concisely described.

Thank you for your suggestions. The format of the paper was discussed with the editor prior to submission.

3) When some abbreviations were mentioned at the first time, they did not follow the full name of the phrase, for example "COGs" first appeared in the Table 1, but until page 6, the authors mentioned "Clusters of Orhologous Groups (COGs)". Furthermore, the "Orhologous" should be "Orthologous". Another example, line 100, the EC number, what does "EC" stand for?

Corrections were made, and full name was used prior to the introduction of abbreviations.

4) Figure 1A might be better shown as a pie plot.

Figure 1 was edited, as suggested.

5) Page 14, the S6 Text can't be found.

We apologize for that mistake. The Supplementary file “Document S1” has now be appropriately presented in the manuscript. Change was made with a similar mistake, regarding supplementary Table 1.

Round 2

Reviewer 3 Report

I am satisfied with the revision. No further comments.